# Smad3 Phospho-Isoform Signaling in Nonalcoholic Steatohepatitis

**DOI:** 10.3390/ijms23116270

**Published:** 2022-06-03

**Authors:** Takashi Yamaguchi, Katsunori Yoshida, Miki Murata, Kanehiko Suwa, Koichi Tsuneyama, Koichi Matsuzaki, Makoto Naganuma

**Affiliations:** 1Department of Gastroenterology and Hepatology, Kansai Medical University, 2-5-1 Shin-machi, Hirakata, Osaka 573-1010, Japan; yoshidka@hirakata.kmu.ac.jp (K.Y.); muratami@takii.kmu.ac.jp (M.M.); suwakan@hirakata.kmu.ac.jp (K.S.); matsuzak@takii.kmu.ac.jp (K.M.); naganuma@hirakata.kmu.ac.jp (M.N.); 2Department of Pathology & Laboratory Medicine, Institute of Biomedical Sciences, Tokushima University Graduate School, 3-18-15 Kuramoto, Tokushima 770-8503, Japan; koichi.tsuneyama@gmail.com

**Keywords:** nonalcoholic steatohepatitis, Smad, transforming growth factor-β, hepatocellular carcinoma

## Abstract

Nonalcoholic fatty liver disease (NAFLD) is characterized by hepatic steatosis with insulin resistance, oxidative stress, lipotoxicity, adipokine secretion by fat cells, endotoxins (lipopolysaccharides) released by gut microbiota, and endoplasmic reticulum stress. Together, these factors promote NAFLD progression from steatosis to nonalcoholic steatohepatitis (NASH), fibrosis, and eventually end-stage liver diseases in a proportion of cases. Hepatic fibrosis and carcinogenesis often progress together, sharing inflammatory pathways. However, NASH can lead to hepatocarcinogenesis with minimal inflammation or fibrosis. In such instances, insulin resistance, oxidative stress, and lipotoxicity can directly lead to liver carcinogenesis through genetic and epigenetic alterations. Transforming growth factor (TGF)-β signaling is implicated in hepatic fibrogenesis and carcinogenesis. TGF-β type I receptor (TβRI) and activated-Ras/c-Jun-N-terminal kinase (JNK) differentially phosphorylate the mediator Smad3 to create two phospho-isoforms: C-terminally phosphorylated Smad3 (pSmad3C) and linker-phosphorylated Smad3 (pSmad3L). TβRI/pSmad3C signaling terminates cell proliferation, while constitutive Ras activation and JNK-mediated pSmad3L promote hepatocyte proliferation and carcinogenesis. The pSmad3L signaling pathway also antagonizes cytostatic pSmad3C signaling. This review addresses TGF-β/Smad signaling in hepatic carcinogenesis complicating NASH. We also discuss Smad phospho-isoforms as biomarkers predicting HCC in NASH patients with or without cirrhosis.

## 1. Introduction

Nonalcoholic fatty liver disease (NAFLD) has become the most prevalent cause of chronic liver disease worldwide [1]. In Japan, there are more than 20 million NAFLD patients, and case numbers are expected to increase [2]. Recent therapeutic advances against hepatitis B and C viruses (HBV and HCV) have reduced hepatocellular carcinoma (HCC) occurrence related to these infections, while HCC related to nonalcoholic steatohepatitis (NASH) is increasing [3]. In Japan, HCV-related HCC now accounts for 67% of all HCC, followed by HCC related to HBV at 16%, while 15.8% of HCC patients have nonviral liver disease [4]. Among nonviral chronic liver diseases, NASH has drawn particular attention over the last decade because of its association with HCC. Typically, HCC arises through cirrhosis complicating prolonged chronic hepatic inflammation [5]. In patients with cirrhosis, proliferation occurs as a compensatory regenerative response to progressive hepatocyte injury [6]. Cirrhosis has been considered a prerequisite for HCC development, and these disease processes share signaling pathways [7]. In contrast, NASH can promote hepatocarcinogenesis in the absence or near-absence of liver fibrosis [8,9,10,11,12]. In NASH, liver fibrosis and HCC are not inexorably linked, with approximately 35% of HCC developing in non-cirrhotic livers [13]; this raises the question of which molecular pathways are common to inflammation, fibrosis, and HCC and which are specific to each. How HCC develops under these disparate conditions remains unclear. In response to this pathogenetic complexity, research has increasingly focused on molecular mechanisms [14].

The pathophysiology underlying NAFLD is complex and unresolved. Tilg et al. proposed the Multiple Parallel Hit hypothesis in which lipotoxicity, adipokine secretion, endotoxins such as lipopolysaccharide (LPS) being released by gut microorganisms, and stress involving the endoplasmic reticulum (ER) act in parallel to promote NAFLD progression from steatosis to NASH, fibrosis, and eventually end-stage liver diseases [15]. This multiple-hit pathogenesis results in lipid accumulation within hepatocytes, with consequent oxidative stress, lipid peroxidation, adipokine signaling, and pro-inflammatory cytokine expression [16]. These inflammatory, fibrogenic, and oncogenic pathways are likely modulated by genetic and epigenetic mechanisms, metabolic and endocrine signaling, immunologic influences, and interactions with intestinal flora [17,18,19,20,21]. Chronic inflammation can cause liver fibrosis and damage DNA in regenerative hepatocytes. This fibrosis and inflammation significantly increase the likelihood of genetic alterations that promote HCC development. On the other hand, insulin resistance, oxidative stress, and lipotoxicity themselves can promote genetic and epigenetic alterations that are not accompanied by chronic inflammation or progressive fibrosis [22,23].

Transforming growth factor (TGF)-β is a cytokine involved in pathologic processes such as hepatic fibrogenesis and carcinogenesis [24,25]. Signal transporters designated Smads, which possess MH1, MH2, and linker domains, convey signals from cell-surface receptors for TGF-β superfamily members to the cell nucleus [26]. Recent studies prove that these TGF-β signaling mediators are tightly controlled by domain-specific phosphorylation, which regulates their subcellular localization, transcriptional response, and component stability [27]. TGF-β binds to type II cell-surface receptors, leading to activation of type I receptors (TβRI). Once activated, TβRI phosphorylates Smad3 at the C-terminal region, producing an isoform designated pSmad3C [28]. In contrast, constitutive Ras activation caused by various mutations increase phosphorylation of a linker region of the Smad to produce a pSmad3L isoform. Such Ras activation also decreases TGF-β-dependent cytostatic pSmad3C function [29]. In addition, Ras-related kinases such as c-Jun N-terminal kinase (JNK) phosphorylate the linker domain of Smad3 [30]. To sum up, TβRI and activated-Ras/JNK act upon Smad3 to produce two different phospho-isoforms: C-terminally phosphorylated Smad3 (pSmad3C) and Smad3 with linker phosphorylation (pSmad3L) [31].

TGF-β-dependent pSmad3C signals impede cell-cycle progression by activating transcription of p15^INK4B^ and p21^CIP1^, while de-activating c-Myc gene transcription [32,33,34]. Thus, pSmad3C-mediated TGF-β signaling is cytostatic and tumor-suppressive [35,36,37]. On the other hand, activated-Ras/JNK-mediated pSmad3L up-regulates transcription of c-Myc, which promotes hepatocytic proliferation leading to liver cancer [38]. In addition, pSmad3L promotes cell invasion and migration by inducing matrix metalloproteinase 2/9 and plasminogen activator inhibitor type 1 [39] (Figure 1). To make matters worse, pSmad3L-induced hepatocyte proliferation is associated with suppressed cytostatic pSmad3C signaling [29].

Overall, the mechanisms of HCC development in NAFLD remain uncertain and require in-depth investigation at the molecular level. Furthermore, the active components and molecular mechanisms of TGF-β/Smad signaling are incompletely understood. Again, further studies are needed. This review addresses TGF-β/Smad signaling implicated in hepatic carcinogenesis and fibrogenesis associated with NASH. We also discuss Smad phospho-isoforms as potential biomarkers predicting HCC in NASH patients with and without cirrhosis.

## 2. Differential Patterns of Carcinogenesis in NASH: A Carcinogenic Sequence from Chronic Inflammation via Progressive Fibrosis versus Carcinogenesis Requiring No Inflammation or Progressive Fibrosis

While HCC is typically accompanied by cirrhosis or severe fibrosis, a growing number of NAFLD patients without cirrhosis or advanced fibrosis have been found to develop HCC. Among individuals without cirrhosis, patients with NASH are at a higher risk of HCC than those with other liver diseases [13]. In NAFLD, HCC can develop at any stage of hepatic fibrosis [40,41], with 20% to 50% of NAFLD-related HCC occurring in patients without cirrhosis. Among 500 patients with HCC with various etiologies, only 43% had cirrhosis [42]. In particular, cirrhosis was present in only 23% of NAFLD-related and 21% of NASH-related HCC cases [42]. In an Italian multicenter cohort of 756 patients with HCC, Piscaglia and colleagues reported that 46.2% of NAFLD-associated HCC occurred on a non-cirrhotic background, in contrast to 97.2% of HCV-associated HCC occurring in cirrhotic livers [12]. Similar results were reported in a German and a Japanese study, in which 41.7% and 49% of NAFLD-associated HCC cases arose in a non-cirrhotic background, respectively [9,10]. Interestingly, there is a clear sexual dimorphism in HCC, with men developing the disease twice as often as women [43]. Estrogen can protect hepatocytes from malignant transformation via downregulation of the IL-6 release from Kupffer cells [44]. Recently, sex differences in hepato-carcinogenesis were extensively explored in the functional signatures of differentially expressed genes [45], expression-level characteristic loci (eQTLs) [46], and cancer driver genes [47], strongly suggesting that male and female HCCs may be biologically different, and the carcinogenic process may differ [48].

Almost all the possible features of NAFLD, including fat deposition, oxidative injury, NASH, liver fibrosis, and cirrhosis, correlate with increased risk of HCC, and provide fertile ground for advancement of these cancers. This is largely driven by both lipotoxicity and insulin resistance, ultimately leading to increased fibrogenesis, inflammation, and abnormal cellular proliferation as well as decreases in apoptotic cell death, necroptosis, and autophagy. Importantly, however, the acute injury–inflammation–fibrosis–cirrhosis–HCC paradigm does not establish a necessary causal link between these processes and HCC, but only an association. As mentioned above, some patients may develop HCC without the occurrence of cirrhosis. Additionally, some studies demonstrate a role of oxidative stress in promoting HCC carcinogenesis in NAFLD, with or without the involvement of inflammation/fibrosis/cirrhosis [49,50,51,52]. Recently, an inverse relationship was found between the extent of fibrosis and tumor size, with non-cirrhosis patients presenting with the largest tumors, suggesting underlying differences in the oncogenic pathway [53]. Mechanisms that underpin hepatocarcinogenesis are complex and involve multiple insults and contributions from genetic modifiers that influence disease severity and progression.

## 3. Chronic Inflammation Is Triggered via a Complex Pathway Leading from Steatosis to NASH to Fibrogenesis

Following a stage of hepatic steatosis, NAFLD may progress to the more complicated state designated NASH. With continuing identification of additional factors that contribute to the development of NASH and elucidation of the many ways in which these factors interact, pathogenesis now is understood to be extremely complex. Though the exact nature of “hits” that follow insulin resistance has not been fully elucidated, evidence has accumulated to suggest probable mechanisms underlying progression from steatosis to NASH, in which oxidative stress-induced cascades involving adipokine secretion and cytokine activation appear to initiate inflammation.

### 3.1. Directly or Indirectly, Oxidative Stress Stimulates Liver Fibrosis through Increases in Pro-Inflammatory Cytokines and Lipotoxicity

In the liver, oxidative stress plays an essential role as an initial response to hepatic and extrahepatic injury. Processes producing ROS include hepatic metabolic responses that are enhanced in obesity such as fatty acid oxidation, ER stress, inflammation, and induction of ROS-producing NAD(P)H oxidases. ROS and aldehydes, secondary products of the oxidation reaction, can cause hepatic stellate cells (HSC) to adopt a myofibroblast (MFB) phenotype capable of producing collagens and other extracellular matrix (ECM) proteins leading to liver fibrosis. These include collagen I (COL I), COL III, COL IV, fibronectin, and α-smooth muscle actin (α-SMA). In addition, ROS alongside products of lipid peroxidation can induce the release of cytokines such as tumor necrosis factor (TNF)-α, interleukin (IL)-1β, and IL6 which participate importantly in inflammation and also induce expression of TNF receptor-1 (TNFR1) [54]. ROS also activate nuclear factor-κB (NF-κB), a redox-sensitive transcription factor that promotes TNF-α expression. All of these alterations may contribute to liver inflammation in NAFLD. Thus, oxidative stress leads to death of hepatocytes and activation of inflammatory pathways, including expression of the pro-inflammatory cytokine TNF, promoting advanced fibrosis and cirrhosis.

Fatty liver is characterized by hepatic steatosis with chronic substrate overload leading to lipotoxicity, an important contributor to NASH. Oxidative stress involving the ER leads to upregulation of lipogenic sterols, resulting in hepatic steatosis. ROS trigger lipid peroxidation, which leads to dissemination of inflammatory cytokines and ultimately cell death. Pro-inflammatory cytokines have been shown to activate JNK through inhibition of mitogen-activated protein kinase (MAPK) phosphatases by ROS. Studies of obese patients with NASH have confirmed elevated hepatic JNK activity. Biologically active lipid peroxidation products and cytokines act together to promote the diverse pathology of NAFLD by inducing inflammation and fibrosis, eventually culminating in end-stage liver diseases including cirrhosis and HCC. Together, oxidative stress and lipotoxicity drive the progression of NASH and fibrosis to end-stage disease [55].

### 3.2. Cytokines Derived from Adipose Tissue Mediate Chronic Inflammation Leading to Insulin Resistance in NASH

Obesity denotes excessive accumulation of adipose tissue exceeding physiologic storage capacity, resulting in ectopic fat deposition that alters metabolic, inflammatory, and immunologic pathways [56]. Adipose tissue secretes several important cytokines that regulate low-grade inflammation; adiponectin and leptin are particularly important [57]. Adiponectin has a strong inverse association with pro-inflammatory cytokines such as TNFα and IL6 [58], which are elevated in NASH. On the other hand, pro-inflammatory TNF-α, IL-1β, IL6, and endotoxin favor the secretion of leptin, which exerts potentially harmful effects on insulin resistance and inflammatory cascades [59]. As a consequence, excessive adipose tissue in obesity is associated with chronic inflammation and insulin resistance as a result of infiltration by activated macrophages and T cells.

How this confluence of an inflammatory microenvironment, aberrant metabolism, and ongoing liver regeneration contributes to DNA instability and cancer remains poorly understood, although tumorigenic signaling was found to be amplified in HCC by over 28,000 mutations [60]. Chronic cell injury triggers the secretion of significant amounts of pro-inflammatory molecules including IL-1, IL6, TNF-α, and lymphotoxin-β, all facilitating HCC development [61]. Mouse studies have explored local intrahepatic chronic inflammatory responses in hepatocarcinogenesis in the context of NASH, finding that TNF derived from inflammatory liver macrophages is crucial in the development of NASH and steatohepatitic HCC in major urinary protein (MUP)-urokinase plasminogen activator (uPA) mice fed a high-fat diet (HFD) [62]. This effect occurs through a transient ER stress response that enhances lipogenesis and worsens hepatic steatosis [63,64,65]. This inflammatory response leads to fibrosis and DNA damage, resulting in hepatocarcinogenesis.

### 3.3. Inflammation in the Absence of Pathogens Leads to Liver Injury and Fibrogenesis in NASH

Kupffer cells (KC) appear to act importantly in the initiation and progression of inflammation and fibrosis [66]. A study using a NASH mouse model demonstrated the importance of TNF-α signaling by hepatic KC in the development of NASH. Further, liver biopsy specimens from patients with NAFLD showed more numerous KC than specimens from controls. Damaged hepatocytes activate KC through cell stress pathways such as JNK and by the release of damage-associated molecular patterns (DAMP), including nuclear and mitochondrial DNA, uric acid, and purine nucleotides; these promote inflammation via TNF-α, NF-κB, and Toll-like receptor (TLR) signaling activation [67,68]. Pathogen-associated molecular patterns (PAMP), which include bacterial products such as LPS, also are important in NAFLD and NASH liver injury. DAMP and PAMP bind to pattern-recognition receptors (PRR), including TLR, triggering a local inflammatory response mediated by cytokines such as TNF-α and IL6 [69,70]. Hepatic cells that express TLR include KC, HSC, biliary epithelial cells, and sinusoidal endothelial cells; the types of TLR most well-studied in NASH are TLR2, TLR4, and TLR9 [71]. TLR-deficient mice exhibited decreased steatosis and inflammation despite a HFD. In early NAFLD, PAMP are altered among gut flora, while increased intestinal translocation of bacteria and toxins can activate KC, which secrete TGF-β, TNF-α, pro-inflammatory cytokines such as CCL2, and ROS, in addition to activating inflammasomes. Most patients with NAFLD and NASH were found to have aberrant overgrowth of intestinal bacteria as well as detectable LPS in portal blood; the latter stimulate TNF-α production by KC through enhancement of TLR4 signaling. This LPS-stimulated TLR4 signaling in both KC and HSC, together with enhanced sensitivity to TGFβ, favors the progression of liver fibrosis [72]. Intestinal permeability, aberrant overgrowth of intestinal bacteria, and inflow of PAMP via the gut–liver axis directly influence NAFLD/NASH status by stimulating the liver’s innate immune system.

Regulation of TLR-mediated inflammatory responses is found to protect against HCC. KC express TLR4, and the binding of LPS results in activation of NF-κB, MAPK, extracellular signal-regulated kinase (ERK)1, p38, JNK, and interferon regulatory factor (IRF)3 [73]. Inflammation-induced suppression of cytotoxic CD8+ T-lymphocyte activation decreases anti-tumor host surveillance [6]. Circulating levels of LPS are increased in animal models of HCC, while prolonged treatment with low-dose LPS significantly increases HCC development [74]. Indeed, the interaction between LPS and TLR4 is crucial to the initiation and promotion of hepatocarcinogenesis through inflammation, chronic liver injury, and fibrosis [75]. A mechanism by which gut bacteria may contribute to hepatocarcinogenesis in addition to LPS interactions is the modulation of the bile acid metabolism [76]. Based on an experimental model of NASH, Yamada et al. suggested that microbiota in the conversion of primary to secondary bile acids by intestinal flora contributes importantly to HCC development [77].

### 3.4. During Chronic Inflammation-Mediated NASH Pathogenesis, TGF-β Signaling Prompts Conversion of HSC to MFB

Extensive experimental data show the importance of HSC activation in fibrotic processes [78]. A key step in fibrogenesis is the conversion of HSC into MFB, while depositing ECM. HSC activation is triggered by multiple mediators secreted by damaged hepatocytes, activated macrophages, and aggregated platelets. Among HSC-activating factors, TGF-β1 is a key molecule regulating MFB function [79]. In chronic liver disease, MFB persist, proliferate, and then migrate, as they continuously deposit ECM which replaces hepatic parenchyma. Further, a recent study suggested the possibility of an interplay between Sonic Hedgehog proteins (Shh) and TGF-β1 in hepatic inflammatory reactions. Secreted Shh may activate TGF-β1 with subsequent activation of HSC, contributing to the progression of NASH in humans [80].

Excessive ECM deposition disrupting normal liver architecture during the fibrogenic process compromises blood flow, interfering with the oxygen supply to the liver parenchyma [81]. Hypoxia triggers activation of specific signaling pathways [82] such as the phosphoinositide 3 kinase (PI3K)-Akt and MAPK pathways, and upregulates angiogenic factors including vascular endothelial growth factor (VEGF) [74] and hypoxia-inducible factor 1 (HIF-1) [83]. Investigating later events, Zheng et al. showed that type 1 collagen accumulation promotes HCC cell proliferation by regulating the β1/FAK integrin pathway in murine models of NAFLD/NASH [84].

### 3.5. Chronic Inflammation Shifts Hepatocytic Smad3 Phospho-Isoform Signaling from Tumor Suppression to Carcinogenesis, Increasing Risk of HCC

Among the pro-inflammatory cytokines, TNF-α is one of the best-characterized cytokines participating in hepatocarcinogenesis, activating JNK signaling pathways [85]. In experiments with TNFRI^−/−^ mice, TNFRI knockout failed to prevent diethylnitrosamine (DEN)-induced HCC, but did suppress development of HFD-promoted DEN-induced HCC, lymphotoxin α/β-overexpression-induced HCC, and hepatocyte-specific TGF-β-activated kinase 1 (TAK1)-deletion-induced HCC [54,86,87,88].

Hepatic JNK activity has been found to be elevated in obese patients with NASH [89,90]. The most compelling evidence for a role of JNK in cancer initiation comes from studies of HCC development in both animal models and human observations. JNK is also one of the most investigated signal transducers with respect to obesity and insulin resistance, as well as the molecular mechanisms linking these two conditions [91]. JNK1 and JNK2 isoforms are expressed in the liver. Mice lacking JNK1, mice lacking JNK2 and heterozygous for JNK1 loss of function, and mice lacking both JNK1 and JNK2 in their livers are largely protected from steatosis under conditions that model diet-induced obesity [92,93,94,95]. These findings indicate that the blockade of JNK may prevent the development of steatosis via both direct and indirect mechanisms [91]. Furthermore, JNK1 activity in hematopoietic cells was implicated in the development of NASH in mice fed a choline-deficient L-amino acid-defined diet, suggesting a major role for JNK1 in the progression of steatohepatitis [96]. Finally, sustained JNK1 activity in hepatocytes [97] and JNK activity in myeloid cells [98] were found to promote chemically induced HCC.

When JNK signaling is active, the involved proteins phosphorylate and activate a wide range of transcription factors and proteins that affect cell proliferation and differentiation, and sometimes promote malignant transformation. Thus, JNK signaling is implicated in a wide range of oncogenic pathways leading to the development of HCC [99], lung cancers [100], prostate cancers [101], and brain tumors [102] via activation of several transcription factors. JNK activates activator protein (AP)-1 [103], which promotes the expression of cyclin D and initiates the G0-to-G1 transition in the cell cycle. TGF-β also activates non-Smad signaling pathways involving TAK1 and JNK [104]. In addition to the modulation of transcriptional responses, the linker phosphorylation of cytoplasmic Smad proteins is critical for the integration of JNK signaling with the TGF-β pathway. Importantly, linker phosphorylation of Smad3 permits translocation of the Smad into the nucleus, where further consequences of JNK signaling proceed [31,105,106,107,108]. Thus, JNK simultaneously activates linker-phosphorylated Smad and nuclear transcription factors binding the Smad complex, with both changes usually occurring in parallel. As a consequence, the linker-phosphorylated Smad pathway is difficult to assess in isolation. To address this problem, we produced domain-specific antibodies (Abs) able to distinguish between the phosphorylated linker region and the C-terminal region of Smad3. These Abs allowed us to determine phosphorylation sites of Smad3, as well as the cellular location of Smad3 in liver tissue in order to understand phospho-Smad3 signaling regulation in hepatocytes better.

We previously immunostained serial sections of liver biopsy specimens from patients with chronic HCV-related liver disease using anti-pSmad3L and anti-pSmad3C Abs [109,110]. Immunostaining of normal livers with an Ab specific to pSmad3C showed slight phosphorylation of Smad3C throughout the liver in contrast to minimal phosphorylation of Smad3L. In cirrhosis and HCC, pSmad3C is less plentiful than in chronic hepatitis C, while pSmad3L gradually increases as liver disease progresses. These observations indicate that hepatocytic Smad3 phospho-isoform signaling can shift from tumor-suppressive pSmad3C signaling to carcinogenic pSmad3L signaling during the progression of chronic liver disease. Synergy between chronic inflammation and persistent HCV infection leads to the accumulation of a molecular condition favoring malignant transformation of hepatocytes. Similarly to that observed in HCV-infected livers, phosphorylation of Smad3L was greater in NASH livers with advanced fibrosis than in NASH livers with mild fibrosis. In contrast, phosphorylation at Smad3C was reduced in highly fibrotic NASH livers compared to in mildly fibrotic NASH livers. Furthermore, HCC developed in most patients with severe fibrosis, in whom hepatocytic pSmad3L was far more abundant, whereas pSmad3C was limited [111]. These observations indicated that, during inflammation-mediated hepatic carcinogenesis, pro-inflammatory conditions such as TNF-α-activated JNK signaling shift hepatocytic Smad3 phospho-isoform signals from those favoring tumor suppression to those promoting carcinogenesis.

## 4. Inflammation-Independent Process of Hepatocarcinogenesis

### 4.1. Oxidative Stress Associated with Insulin Resistance and Lipotoxicity Can Mediate Carcinogenesis Even without Inflammation or Fibrogenesis

Elevated blood glucose has been associated with increased cancer risk in a number of prospective studies [112,113,114,115,116,117,118]. A large collaborative study of over 550,000 patients in six prospectively studied cohorts demonstrated that, independently of body mass index, abnormal glucose metabolism was associated with increased risk of cancer at various sites in both men and women [119]. Hepatic lipid accumulation leads to metabolic reprogramming characterized by a combination of cellular metabolic alterations and accumulation of potentially toxic metabolites favoring hepatic tumorigenesis. Excessive oxidative stress from a disordered lipotoxic metabolism, formation of lipotoxic metabolites, and release of ROS may directly promote liver carcinogenesis.

Patients with NASH were reported to have more hepatic oxidative DNA damage than patients with other liver diseases [120,121]. Moreover, worse oxidative DNA damage was demonstrated in hepatocytes from patients with NASH and HCC than in hepatocytes from patients with NASH but not HCC [121]. Oxidative stress and insulin resistance stimulate signaling via NF-κB and an inhibitor of the kappa light polypeptide gene enhancer in B-cells, kinase beta (IKKβ), increasing survival of damaged hepatocytes [122]. Suppression of immune surveillance and an activated state of hepatocytes and KC caused by oxidative stress contribute further to carcinogenesis.

Although the ROS generated by phagocytic cells represent a first line of defense against pathogens, bystander DNA damage from ROS produced by macrophages and neutrophils recruited in the context of NAFLD likely contributes to the development of HCC, intrahepatic cholangiocarcinoma, and gastrointestinal cancers. Mechanisms involved in oxidative stress-mediated carcinogenesis include altered activity in signaling pathways affecting cell-growth/survival and the development of cancer (e.g., signal transducer and activator of transcription (STAT) 1, STAT3, TNF-α, NF-κB, IL-1, and IFN-γ) [123,124,125]. Oxidative stress also favors the accumulation of oncogenic mutations (e.g., p53, Wnt, Notch, cIAP1, and Yap) via free radical-induced DNA damage, DNA repair inhibition, and telomere shortening, resulting in both genetic and genomic alterations [126,127,128,129]. Oxidative damage to mitochondria interferes with the flow of electrons in the respiratory chain and alters mitochondrial respiratory chain polypeptides and mitochondrial DNA to increase mitochondrial ROS formation, representing a vicious cycle of expanding damage [52].

Recent experimental studies show increased lipogenesis in tumor cells based on an increase in two key up-regulators of lipogenesis, stearoyl-CoA desaturase and fatty acid synthase, as well as the appearance of a visibly lipid-rich phenotype among tumor cells, which were engorged with lipid droplets, as visualized by electron microscopy [130]. Such de novo lipogenesis is a hallmark of cancer. Excesses of triglycerides and FFA suppress the initiation of autophagy through activation of mammalian target of rapamycin (mTOR) and suppression of serine/threonine-protein kinase ULK1 activity, leading to increased hepatic oxidative stress [131,132].

Sestrins are multifunctional proteins that activate adenosine monophosphate-activated protein kinase (AMPK) and mTOR kinase complex 2 (mTORC2) while inhibiting mTORC1 [133]. Leucine may influence regulation of mTORC1 by sestrin1 (Sesn1) and Sesn2 [134,135]. Sesn1 and Sesn2 also activate nuclear factor erythroid 2-related factor 2 transcriptional activity through autophagic degradation of Kelch-like ECH-associated protein 1 in response to oxidative stress [136]. Sesn3 regulates hepatic lipid homeostasis by increasing fatty acid oxidation through activation of AMPK and decreasing lipogenesis mediated by AMPK activation and mTORC1 suppression [137]. Fatty acid synthesis provides tumor cells with metabolic intermediates essential for synthesis of their membrane lipid bilayer, for energy storage as triglycerides, and for signaling molecules [138]. Enhanced activity of JNK in NASH has been attributed to saturated FFA, which were shown to activate JNK pathways [139]. JNK acts as a negative regulator of peroxisome proliferator-activated receptor α activity and fibroblast growth factor 21 expression in hepatocytes, acting via the induction of nuclear receptor corepressor 1. Thus, JNK reduces hepatic fatty acid oxidation and ketogenesis while promoting hepatic steatosis and insulin resistance during diet-induced obesity [91,94,95].

IL6 expression was reported to be increased in patients with chronic hepatitis, for example following HBV or HCV infection, and also in NASH [140,141,142]. IL6 correlates positively with risk of HCC in patients with chronic hepatitis. Further, cytokine directly contributes to genomic instability in hepatocytes while favoring development of cancer during liver regeneration. However, numerous studies in animal models of chronic hepatitis indicate that IL6 signaling can sometimes ameliorate liver injury and fibrosis. Moreover, recent evidence from both clinical studies and animal models suggests a clear linkage between reduced IL6 signaling and obesity [143,144,145], which is now considered a major independent risk factor in HCC development. In an elegant study, Grohmann et al. demonstrated how obesity-associated hepatic oxidative stress can independently contribute to pathogenesis of NASH, fibrosis, and HCC via STAT-1 and STAT-3 signaling [124]. While STAT-1 signaling was responsible for recruitment of activated cytotoxic T cells and ensuing NASH and fibrosis, STAT-1 was not essential for HCC to occur. Rather, T-cell protein tyrosine phosphatase inactivation was found to promote HCC in obesity via STAT-3, independently of T-cell recruitment, NASH, or fibrosis. Such results shed light on mechanisms that may underlie the growing incidence of HCC in non-cirrhotic livers in the setting of NAFLD [23].

### 4.2. The Mutational Landscape of NASH-HCC

Development and progression of HCC is a multistep event governed by accumulation of genetic and epigenetic changes favoring activation of oncogenes that inactivate tumor suppressor genes and disrupt fundamental cellular processes. Several signaling pathways are implicated in hepatocarcinogenesis, including dysregulation of proliferation, differentiation, inflammation, neoangiogenesis, and p53 tumor suppression. During early liver cancer development in mouse NAFLD models, oncogene activation led to DNA damage and chromosomal instability [146]. Ras/MAPK, Wnt/b-catenin, TGF-β/Smad, and Rho/actin pathways act together to influence other complex networks of interactions enhancing tumor cell proliferation, differentiation, epithelial-to-mesenchymal transition, and tumor invasiveness [147,148,149,150,151]. Ras/MAPK signaling is among the pathways most implicated in HCC, despite the absence of activating Ras mutations [152,153,154]. A critical physiologic counterpart of Ras is Ras-association domain-containing protein 1 (RASSF1a), which is an established tumor suppressor embedded within an intricate regulatory network [155]. RASSF1a has gained importance as a frequent epigenetically silenced target of aberrant hypermethylation in diverse types of cancer [156], including HCC [153,157].

Pinyol and colleagues explored the mutational landscape of NASH-HCC, finding TERT promoter (56%), CTNNB1 (28%), TP53 (18%) and activin A receptor type 2A (ACVR2A; 10%) to be the most frequently mutated genes. They found higher rates of ACVR2A mutations among NASH-HCC than among virus- or alcohol-related HCC [158]. ACVR2A is a cytokine receptor involved in cell differentiation and proliferation, reported to be mutated in microsatellite-unstable colorectal cancers, and its downregulation is associated with poor outcomes [159]. Mutation of ACVR2A, a potential tumor suppressor, and the presence of a novel mutational signature characterize NASH-related HCC. Importantly, ACVR2A mutations appear to be unassociated with cirrhosis [158]. Finally, increased mitochondrial activity was reported to increase concentrations of ROS and subsequent DNA damage; both were reported as initial carcinogenic steps. The PI3K–AKT–mTOR pathway is thought to be activated in most tumors arising from upstream mutations, such as AKT, PTEN, and MYC, but these tumors are also influenced by independent oncogenic pathways such as the MAPK–ERK pathway [160].

### 4.3. Even without Chronic Inflammation and Progressive Fibrosis, Inflammation-Independent Hepatic Carcinogenesis May Arise from Genetic or Epigenetic Alterations

Constitutively activated Ras induces ongoing phosphorylation of Smad3 at its linker region. Highly phosphorylated Smad3L is likely to impair sensitivity to growth inhibition by pSmad3C in tumor cells [161]. Thus, mutations in key pathway components lead to sustained linker phosphorylation of Smad3. Further, a frequent hallmark of tumor cells is overexpression and/or amplification of growth factor receptors on cell surfaces, resulting in aberrant constitutive linker phosphorylation of Smad3 in the absence of pro-inflammatory cytokines. Our previous study showed that NASH patients with abundant pSmad3L in the liver can develop HCC, even when fibrosis is minimal [111]. Since most of these patients display decreased inflammatory activity, progression of disease appears to have lost dependence of inflammation, and the oncogenic pSmad3L signal persists because of acquired genetic or epigenetic changes in hepatocytes.

## 5. Analysis of Phospho-Smad Signaling Is Useful for Targeting High-Risk NASH Patients Who Require Intense Surveillance for HCC

Increasingly, HCC has been arising from nonviral liver disease, requiring more effective screening for early detection of HCC. According to reports of Japanese annual health checks, 9% to 30% of adults demonstrate evidence of NAFLD according to ultrasonography. Since some 10% to 20% of individuals with NAFLD have NASH; the prevalence of NASH can be estimated at 1% to 3% of the adult Japanese population, representing an extremely large number of cases [162,163,164]. Because of difficulty in accurately defining the population at high risk for HCC in NASH, NASH-HCC is unfortunately often diagnosed at a later tumor stage [12].

Stratification by fibrosis score may help to define the subgroup of non-cirrhotic NASH most likely to benefit from screening. Previous studies suggest that noninvasive or minimally invasive biomarkers including platelet counts, serum concentration of type IV collagen 7s domain (7s collagen), Wisteria floribunda agglutinin-positive Mac-2 binding protein, fibrosis-4 (FIB-4) index evaluation, and liver stiffness determined by imaging technologies are possible methods for estimating the degree of liver fibrosis in patients with NAFLD [165,166,167,168,169]. Considering the increased risk of HCC in patients with NAFLD who also have obesity and type 2 diabetes, some guidelines suggest individualized HCC surveillance or further clinical risk stratification in such patients [162,170]. However, no specific surveillance recommendations exist for patients with non-cirrhotic NAFLD. How can we better stratify patients with NASH who are at high risk for HCC using reliably specific biomarkers? Activation of the JNK pathway by stress signals associated with persistent viral infection and chronic inflammation has been studied in much detail, but how closely JNK is tied to cancer risk in patients with chronic liver disease associated with HCV or NASH remains uncertain. Our previous study indicated that hepatocytic Smad3 phospho-isoform signaling shifts from tumor-suppressive pSmad3C/ p21^WAF1^ signaling to carcinogenic pSmad3L/c-Myc signaling during HCV-related chronic liver disease progression [109]. This observation suggests that synergy between chronic inflammation and persistent HCV infection leads to multiple events contributing to malignant transformation of hepatocytes. In a recent study, we found that in NASH-related HCC, pSmad3L increased as fibrosis progressed. HCC was especially likely to occur in our group with high hepatic pSmad3L, while high pSmad3C signaling appeared to protect from HCC [111]. As in HCV-related chronic liver disease, some NASH patients showed inflammation leading to fibrosis and carcinogenesis. As mentioned earlier, ROS, oxidative stress, lipotoxicity exacerbated by ROS, PAMP and DAMP such as LPS, adipokines, and insulin resistance all contribute to chronic inflammation in NASH patients. However, unlike patients with HCV-related chronic liver disease, some NASH patients may develop HCC without chronic inflammation or progressive fibrosis. In such situations, direct DNA damage may result from insulin resistance, lipotoxicity, and ROS caused by oxidative stress. Most importantly, whether chronic inflammation or host factors are more at fault, hepatocytic phospho-Smad3 signaling can shift from tumor suppression to carcinogenesis, increasing the risk of hepatocellular carcinoma (Figure 2).

Regardless of the degree of liver fibrosis, we found more abundant pSmad3L in NASH-HCC than in non-HCC patients. In addition, HCC was more likely to occur in patients with high pSmad3L. On the other hand, low pSmad3L is associated with a low rate of cancer development [111]. We found phospho-Smad isoform profiles to be useful biomarkers for predicting early development of HCC in NASH. Such predictive markers should allow us to identify groups of patients with NAFLD and NASH, with or without fibrosis progression, who are at high and low risk of developing hepatocellular carcinoma. Such a profile would allow for evaluation of the effectiveness of interventions aimed at reducing cancer risk in humans.

## 6. Conclusions

This review described some potential molecular mechanisms (Table 1) that contribute to hepatic carcinogenesis with or without fibrogenesis in NASH. The identification of individuals with a high risk of HCC among patients with NASH is essential to improve patient survival. We therefore need to understand the molecular mechanisms of carcinogenesis in this context better, and must identify molecular targets for new preventive and therapeutic approaches in this population. Decreasing pSmad3L may eventually prove to be a new approach to preventing NASH-related HCC.

## Figures and Tables

**Figure 1 ijms-23-06270-f001:**
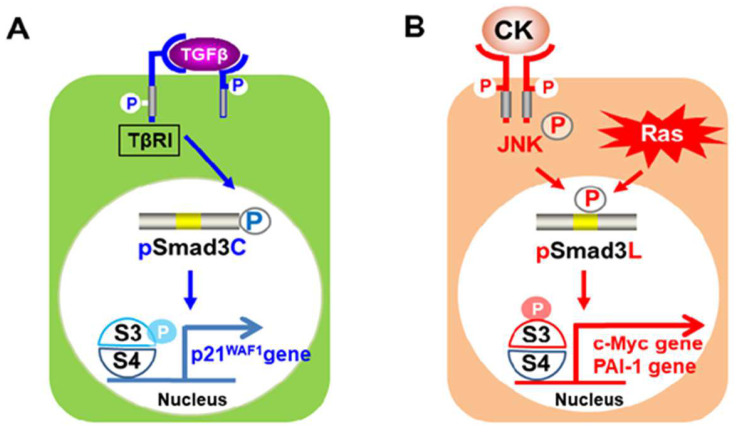
(**A**) Activated transforming growth factor (TGF)-β type I receptor (TβRI) phosphorylates COOH-tail serine residues of Smad3. Phosphorylated Smad3C translocates with Smad4 to suppress cell growth, stimulating the p21^waf1^ promoter. (**B**) Pro-inflammatory cytokines (CK) such as tumor necrosis factor (TNF)-α activate c-Jun N-terminal kinase (JNK) or Ras signaling to phosphorylate the linker region of Smad3. Linker-phosphorylated Smad3 (pSmad3L) translocates with Smad4 to the nucleus and up-regulates c-Myc, stimulating cell proliferation as well as plasminogen activator inhibiter type 1 (PAI-1), promoting cell invasion and migration.

**Figure 2 ijms-23-06270-f002:**
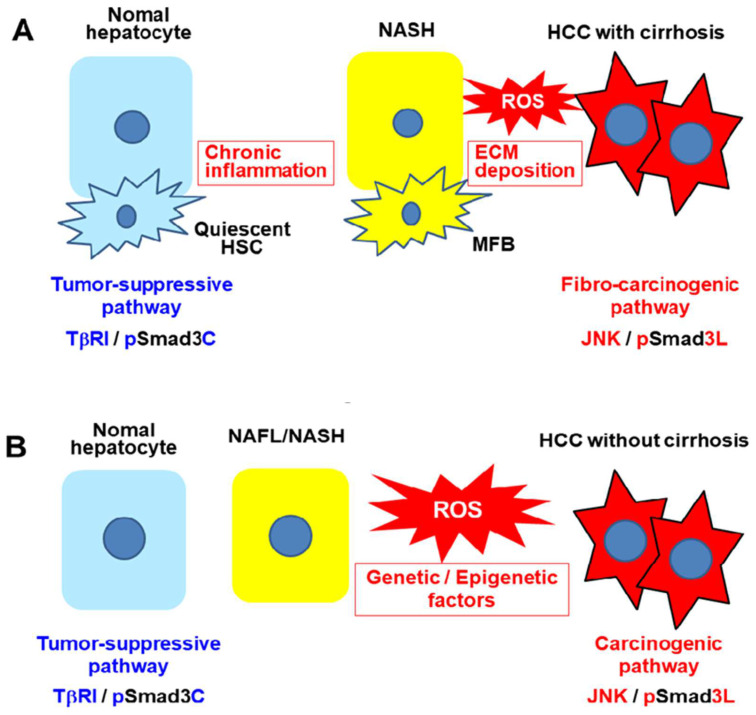
(**A**) In the course of NASH progression, chronic inflammation is induced by oxidative stress in which reactive oxygen species (ROS), lipotoxicity exacerbated by ROS, pathogen-associated molecular patterns (PAMP), and damage-associated molecular patterns (DAMP) such as lipopolysaccharides, adipokines, and insulin resistance promote fibrosis progression to cirrhosis, with shifting of hepatocytic Smad3 phospho-isoform signaling from tumor suppression to carcinogenesis, increasing risk of HCC. ECM, extracellular matrix. (**B**) Even in the absence of chronic inflammation and progressive fibrosis, an inflammation-independent process of hepatic carcinogenesis may occur as a result of genetic or epigenetic alteration. In such situations, direct DNA damage may occur from insulin resistance and lipotoxicity, as well as ROS caused by oxidative stress. Thus, mutations in key pathway components lead to sustained linker phosphorylation of Smad3, impairing sensitivity to growth inhibition by pSmad3C.

**Table 1 ijms-23-06270-t001:** Potential triggers of NASH-related carcinogenesis.

Triggers	Mechanisms	Effect	Reference
Oxidative stress	Pro-inflammatory cytokines	Inflammation	[54]
	TNFR1 expression		
	Lipid peroxidation		
	HSC activation	Fibrosis	[126,127,128,129]
	Oncogenic mutation	DNA damage	
Lipotoxicity	Pro-inflammatory cytokines	Inflammation	[55]
	ER stress	DNA damage	[63,64,65]
Adipose tissue	Adiponectin	Inflammation	[57,58]
	Insulin resistance		
PAMPs	KC and HSC activation	Inflammation	[69,70,72]
		Fibrosis	

PAMPs, pathogen-associated molecular patterns; TNFR1, tumor necrosis factor receptor-1; HSC, hepatic stellate cells; ER, endoplasmic reticulum; KC, Kupffer cells.

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
