# Peer review of "Smad3 Phospho-Isoform Signaling in Nonalcoholic Steatohepatitis"

_ijms, 2022, doi:10.3390/ijms23116270_

Round 1
Reviewer 1 Report
Reviewer comments for ijms-1710035
This study principally reviewed the Smad3 phospho-isoform signaling in nonalcoholic steatohepatitis-derived hepatic carcinogenesis. Furthermore, it was also discussed whether Smad3 phospho-isoforms can be applied as a biomarker for predicting HCC with NASH. The question is clear, and the topic is interesting. However, several major and minor issues still need to be revised and elucidated.
- In the present paper, the content and style of Figures 1 and 2 are a lot similar to the previous paper, which the authors themselves publish [1]. We suggest remaking all the figures to avoid repetition with the published one.
[1] Yamaguchi T, Yoshida K, Murata M, Matsuzaki K. Smad3 phospho-isoform signaling in hepatitis C virus-related chronic liver diseases. World J Gastroenterol 2014; 20(35): 12381-12390 [PMID: 25253939 DOI: 10.3748/wjg.v20.i35.12381]
- Instead of text, using appropriate tables for summarizing will make the manuscript easier to be understood.
- In the abstract portion, LPS is the abbreviation of lipopolysaccharides. Therefore, “lipopolysaccharides or LPS” is an Inappropriate expression.
- In the page 10, 2nd paragraph, line 10, “exits” might be a typing error for “exists”, please check and revise it.
Reviewer 2 Report
Excellent and insightful review of Smad3 signaling in NAFLD.
Reviewer 3 Report
Abstract “ these factors promote NAFLD progression from steatosis to nonalcoholic steatohepatitis (NASH), fibrosis, and eventually end-stage liver diseases” → It should be added “in a proportion of cases”
“Previously, a “two-hit hypothesis” was proposed to explain the mechanisms under-lying initiation and progression of NAFLD” → Authors may be willing to refer to the more recent and credible “multiple-hit” hypothesis Hepatology. 2021 Feb;73(2):833-842.
Both NAFLD and HCC exhibit strong sexual dimorphism (Clin Gastroenterol Hepatol. 2021 Jan;19(1):61-71.e15. Hepatoma Res 2020;6:83. 10.20517/2394-5079.2020.89). Authors may be willing to discuss this feature while highlighting the possible biological grounds for this sex disparity in HCC.
Additionally, it would be important to discuss how an improved understanding of molecular pathogenesis of HCC developing in a NAFLD/NASH milieu may eventually result in implementing precision medicine approaches to NASH diagnosis and management.
Further citations:
Hepatoma Res 2021;7:70.10.20517/2394-5079.2021.74
Metab Target Organ Damage 2022;2:8. http://dx.doi.org/10.20517/mtod.2022.05
Round 2
Reviewer 1 Report
Thank you for your careful revision. A minor error should be addressed.
In Table 1, the full name of the abbreviation is missing. Please add it at the end of the table.
